# Processing of missing data by neural networks

**Marek Śmieja**
marek.smieja@uj.edu.pl

**Łukasz Struski**
lukasz.struski@uj.edu.pl

**Jacek Tabor**
jacek.tabor@uj.edu.pl

**Bartosz Zieliński**
bartosz.zielinski@uj.edu.pl

**Przemysław Spurek**
przemyslaw.spurek@uj.edu.pl

Faculty of Mathematics and Computer Science
Jagiellonian University
Łojasiewicza 6, 30-348 Kraków, Poland

## Abstract

We propose a general, theoretically justified mechanism for processing missing data by neural networks. Our idea is to replace typical neuron's response in the first hidden layer by its expected value. This approach can be applied for various types of networks at minimal cost in their modification. Moreover, in contrast to recent approaches, it does not require complete data for training. Experimental results performed on different types of architectures show that our method gives better results than typical imputation strategies and other methods dedicated for incomplete data.

## 1   Introduction

Learning from incomplete data has been recognized as one of the fundamental challenges in machine learning [1]. Due to the great interest in deep learning in the last decade, it is especially important to establish unified tools for practitioners to process missing data with arbitrary neural networks.

In this paper, we introduce a general, theoretically justified methodology for feeding neural networks with missing data. Our idea is to model the uncertainty on missing attributes by probability density functions, which eliminates the need of direct completion (imputation) by single values. In consequence, every missing data point is identified with parametric density, e.g. GMM, which is trained together with remaining network parameters. To process this probabilistic representation by neural network, we generalize the neuron's response at the first hidden layer by taking its expected value (Section 3). This strategy can be understand as calculating the average neuron's activation over the imputations drawn from missing data density (see Figure 1 for the illustration).

The main advantage of the proposed approach is the ability to train neural network on data sets containing only incomplete samples (without a single fully observable data). This distinguishes our approach from recent models like context encoder [2, 3], denoising autoencoder [4] or modified generative adversarial network [5], which require complete data as an output of the network in training. Moreover, our approach can be applied to various types of neural networks what requires only minimal modification in their architectures. Our main theoretical result shows that this generalization does not lead to loss of information when processing the input (Section 4). Experimental results performed on several types of networks demonstrate practical usefulness of the method (see Section 5 and Figure 2 for sample results) .

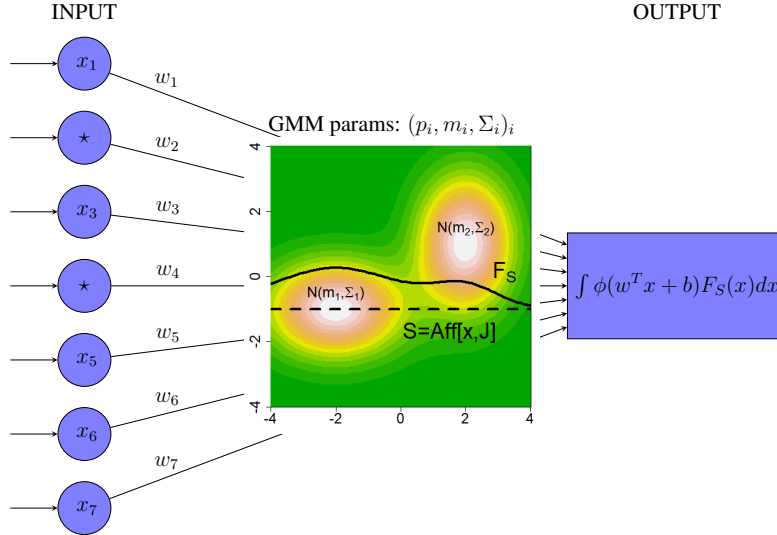

Figure 1: Missing data point $(x, J)$, where $x \in \mathbb{R}^D$ and $J \subset \{1, \ldots, D\}$ denotes absent attributes, is represented as a conditional density $F_S$ (data density restricted to the affine subspace $S = \text{Aff}[x, J]$ identified with $(x, J)$). Instead of calculating the activation function $\phi$ on a single data point (as for complete data points), the first hidden layer computes the expected activation of neurons. Parameters of missing data density $(p_i, \mu_i, \Sigma_i)_i$ are tuned jointly with remaining network parameters.

## 2 Related work

Typical strategy for using machine learning methods with incomplete inputs relies on filling absent attributes based on observable ones [6], e.g. mean or k-NN imputation. One can also train separate models, e.g. neural networks [7], extreme learning machines (ELM) [8], $k$-nearest neighbors [9], etc., for predicting the unobserved features. Iterative filling of missing attributes is one of the most popular technique in this class [10, 11]. Recently, a modified generative adversarial net (GAN) was adapted to fill in absent attributes with realistic values [12]. A supervised imputation, which learns a replacement value for each missing attribute jointly with remaining network parameters, was proposed in [13].

Instead of generating candidates for filling missing attributes, one can build a probabilistic model of incomplete data (under certain assumptions on missing mechanism) [14, 15], which is subsequently fed into particular learning model [16, 17, 18, 19, 20, 21, 22, 23]. Decision function can also be learned based on the visible inputs alone [24, 25], see [26, 27] for SVM and random forest cases. Pelckmans et. al. [28] modeled the expected risk under the uncertainty of the predicted outputs. The authors of [29] designed an algorithm for kernel classification under low-rank assumption, while Goldberg et. al. [30] used matrix completion strategy to solve missing data problem.

The paper [31] used recurrent neural networks with feedback into the input units, which fills absent attributes for the sole purpose of minimizing a learning criterion. By applying the rough set theory, the authors of [32] presented a feedforward neural network which gives an imprecise answer as the result of input data imperfection. Goodfellow et. al. [33] introduced the multi-prediction deep Boltzmann machine, which is capable of solving different inference problems, including classification with missing inputs.

Alternatively, missing data can be processed using the popular context encoder (CE) [2, 3] or modified GAN [5], which were proposed for filling missing regions in natural images. The other possibility would be to use denoising autoencoder [4], which was used e.g. for removing complex patterns like superimposed text from an image. Both approaches, however, require complete data as an output of the network in training phase, which is in contradiction with many real data sets (such us medical ones).

# 3 Layer for processing missing data

In this section, we present our methodology for feeding neural networks with missing data. We show how to represent incomplete data by probability density functions and how to generalize neuron's activation function to process them.

**Missing data representation.** A missing data point is denoted by $(x, J)$, where $x \in \mathbb{R}^D$ and $J \subset \{1, \ldots, D\}$ is a set of attributes with missing values. With each missing point $(x, J)$ we associate the affine subspace consisting of all points which coincide with $x$ on known coordinates $J' = \{1, \ldots, N\} \setminus J$:

$$S = \text{Aff}[x, J] = x + \text{span}(e_J),$$

where $e_J = [e_j]_{j \in J}$ and $e_j$ is $j$-th canonical vector in $\mathbb{R}^D$.

Let us assume that the values at missing attributes come from the unknown $D$-dimensional probability distribution $F$. Then we can model the unobserved values of $(x, J)$ by restricting $F$ to the affine subspace $S = \text{Aff}[x, J]$. In consequence, possible values of incomplete data point $(x, J)$ are described by a conditional density[1] $F_S : S \to \mathbb{R}$ given by (see Figure 1):

$$F_S(x) = \begin{cases} \frac{1}{\int_S F(s)ds} F(x), \text{ for } x \in S, \\ 0, \text{ otherwise.} \end{cases} \quad (1)$$

Notice that $F_S$ is a degenerate density defined on the whole $\mathbb{R}^D$ space[2], which allows to interpret it as a probabilistic representation of missing data point $(x, J)$.

In our approach, we use the mixture of Gaussians (GMM) with diagonal covariance matrices as a missing data density $F$. The choice of diagonal covariance reduces the number of model parameters, which is crucial in high dimensional problems. Clearly, a conditional density for the mixture of Gaussians is a (degenerate) mixture of Gaussians with a support in the subspace. Moreover, we apply an additional regularization in the calculation of conditional density (1) to avoid some artifacts when Gaussian densities are used[3]. This regularization allows to move from typical conditional density given by (1) to marginal density in the limiting case. Precise formulas for a regularized density for GMM with detailed explanations are presented in Supplementary Materials (section 1).

**Generalized neuron's response.** To process probability density functions (representing missing data points) by neural networks, we generalize the neuron's activation function. For a probability density function $F_S$, we define the generalized response (activation) of a neuron $n : \mathbb{R}^D \to \mathbb{R}$ on $F_S$ as the mean output:

$$n(F_S) = E[n(x)|x \sim F_S] = \int n(x) F_S(x) dx.$$

Observe that it is sufficient to generalize neuron's response at the first layer only, while the rest of network architecture can remain unchanged. Basic requirement is the ability of computing expected value with respect to $F_S$. We demonstrate that the generalized response of ReLU and RBF neurons with respect to the mixture of diagonal Gaussians can be calculated efficiently.

Let us recall that the ReLU neuron is given by

$$\text{ReLU}_{w,b}(x) = \max(w^T x + b, 0),$$

where $w \in \mathbb{R}^D$ and $b \in \mathbb{R}$ is the bias. Given 1-dimensional Gaussian density $N(m, \sigma^2)$, we first evaluate $\text{ReLU}[N(m, \sigma^2)]$, where $\text{ReLU} = \max(0, x)$. If we define an auxiliary function:

$$\text{NR}(w) = \text{ReLU}[N(w, 1)],$$

then the generalized response equals:

$$\text{ReLU}[N(m, \sigma^2)] = \sigma \text{NR}(\frac{m}{\sigma}).$$

Elementary calculation gives:

$$\mathrm{NR}(w) = \frac{1}{\sqrt{2\pi}}\exp(-\frac{w^2}{2}) + \frac{w}{2}(1 + \mathrm{erf}(\frac{w}{\sqrt{2}})), \qquad (2)$$

where $\mathrm{erf}(z) = \frac{2}{\sqrt{pi}}\int_0^z \exp(-t^2)dt$.

We proceed with a general case, where an input data point $x$ is generated from the mixture of (degenerate) Gaussians. The following observation shows how to calculate the generalized response of $\mathrm{ReLU}_{w,b}(x)$, where $w \in \mathbb{R}^D, b \in \mathbb{R}$ are neuron weights.

**Theorem 3.1.** *Let* $F = \sum_i p_i N(m_i, \Sigma_i)$ *be the mixture of (possibly degenerate) Gaussians. Given weights* $w = (w_1, \ldots, w_D) \in \mathbb{R}^D, b \in \mathbb{R}$*, we have:*

$$\mathrm{ReLU}_{w,b}(F) = \sum_i p_i \mathrm{NR}\left(\frac{w^T m_i + b}{\sqrt{w^T \Sigma_i w}}\right).$$

*Proof.* If $x \sim N(m, \Sigma)$ then $w^T x + b \sim N(w^T x + b, w^T \Sigma w)$. Consequently, if $x \sim \sum_i p_i N(m_i, \Sigma_i)$, then $w^T x + b \sim \sum_i p_i N(w^T m_i + b, w^T \Sigma_i w)$.

Making use of (2), we get:

$$\mathrm{ReLU}_{w,b}(F) = \int_{\mathbb{R}} \mathrm{ReLU}(x) \sum_i p_i N(w^T m_i + b, w^T \Sigma_i w)(x)dx$$

$$= \sum_i p_i \int_0^\infty x N(w^T m_i + b, w^T \Sigma_i w)(x)dx = \sum_i p_i \mathrm{NR}\left(\frac{w^T m_i + b}{\sqrt{w^T \Sigma_i w}}\right).$$

$\square$

We show the formula for a generalized RBF neuron's activation. Let us recall that RBF function is given by $\mathrm{RBF}_{c,\Gamma}(x) = N(c, \Gamma)(x)$.

**Theorem 3.2.** *Let* $F = \sum_i p_i N(m_i, \Sigma_i)$ *be the mixture of (possibly degenerate) Gaussians and let RBF unit be parametrized by* $N(c, \Gamma)$*. We have:*

$$\mathrm{RBF}_{c,\Gamma}(F) = \sum_{i=1}^k p_i N(m_i - c, \Gamma + \Sigma_i)(0).$$

*Proof.* We have:

$$\mathrm{RBF}_{c,\Gamma}(F) = \int_{\mathbb{R}^D} \mathrm{RBF}_{c,\Gamma}(x)F(x)dx = \sum_{i=1}^k p_i \int_{\mathbb{R}^D} N(c, \Gamma)(x)N(m^i, \Sigma^i)(x)dx$$

$$= \sum_{i=1}^k p_i \langle N(c, \Gamma), N(m^i, \Sigma^i)\rangle = \sum_{i=1}^k p_i N(m_i - c, \Gamma + \Sigma^i)(0). \quad (3)$$

$\square$

**Network architecture.** Adaptation of a given neural network to incomplete data relies on the following steps:

1. *Estimation of missing data density with the use of mixture of diagonal Gaussians.* If data satisfy missing at random assumption (MAR), then we can adapt EM algorithm to estimate incomplete data density with the use of GMM. In more general case, we can let the network to learn optimal parameters of GMM with respect to its cost function[4]. The later case was examined in the experiment.

2. *Generalization of neuron's response.* A missing data point $(x, J)$ is interpreted as the mixture of degenerate Gaussians $F_S$ on $S = \text{Aff}[x, J]$. Thus we need to generalize the activation functions of all neurons in the first hidden layer of the network to process probability measures. In consequence, the response of $n(\cdot)$ on $(x, J)$ is given by $n(F_S)$.

The rest of the architecture does not change, i.e. the modification is only required on the first hidden layer.

Observe that our generalized network can also process classical points, which do not contain any missing values. In this case, generalized neurons reduce to classical ones, because missing data density $F$ is only used to estimate possible values at absent attributes. If all attributes are complete then this density is simply not used. In consequence, if we want to use missing data in testing stage, we need to feed the network with incomplete data in training to fit accurate density model.

# 4 Theoretical analysis

There appears a natural question: how much information we lose using generalized neuron's activation at the first layer? Our main theoretical result shows that our approach does not lead to the lose of information, which justifies our reasoning from a theoretical perspective. For a transparency, we will work with general probability measures instead of density functions. The generalized response of neuron $n : \mathbb{R}^D \to \mathbb{R}$ evaluated on a probability measure $\mu$ is given by:

$$n(\mu) := \int n(x) d\mu(x).$$

The following theorem shows that a neural network with generalized ReLU units is able to identify any two probability measures. The proof is a natural modification of the respective standard proofs of Universal Approximation Property (UAP), and therefore we present only its sketch. Observe that all generalized ReLU return finite values iff a probability measure $\mu$ satisfies the condition

$$\int \|x\| d\mu(x) < \infty. \tag{4}$$

That is the reason why we reduce to such measures in the following theorem.

**Theorem 4.1.** *Let $\mu, \nu$ be probabilistic measures satisfying condition* (4). *If*

$$\text{ReLU}_{w,b}(\mu) = \text{ReLU}_{w,b}(\nu) \text{ for } w \in \mathbb{R}^D, b \in \mathbb{R}, \tag{5}$$

*then $\nu = \mu$.*

*Proof.* Let us fix an arbitrary $w \in \mathbb{R}^D$ and define the set

$$\mathcal{F}_w = \Big\{ p : \mathbb{R} \to \mathbb{R} : \int p(w^T x) d\mu(x) = \int p(w^T x) d\nu(x) \Big\}.$$

Our main step in the proof lies in showing that $\mathcal{F}_w$ contains all continuous bounded functions.

Let $r_i \in \mathbb{R}$ such that $-\infty = r_0 < r_1 < \ldots < r_{l-1} < r_l = \infty$ and $q_i \in \mathbb{R}$ such that $q_0 = q_1 = 0 = q_{l-1} = q_l$, be given. Let $Q : \mathbb{R} \to \mathbb{R}$ be a piecewise linear continuous function which is affine linear on intervals $[r_i, r_{i+1}]$ and such that $Q(r_i) = q_i$. We show that $Q \in \mathcal{F}_w$. Since

$$Q = \sum_{i=1}^{l-1} q_i \cdot T_{r_{i-1}, r_i, r_{i+1}},$$

where the tent-like piecewise linear function $T$ is defined by

$$T_{p_0, p_1, p_2}(r) = \begin{cases} 0 \text{ for } r \leq p_0, \\ \frac{r - p_0}{p_1 - p_0} \text{ for } r \in [p_0, p_1], \\ \frac{p_2 - r}{p_2 - p_1} \text{ for } r \in [p_1, p_2], \\ 0 \text{ for } r \geq p_2, \end{cases}$$

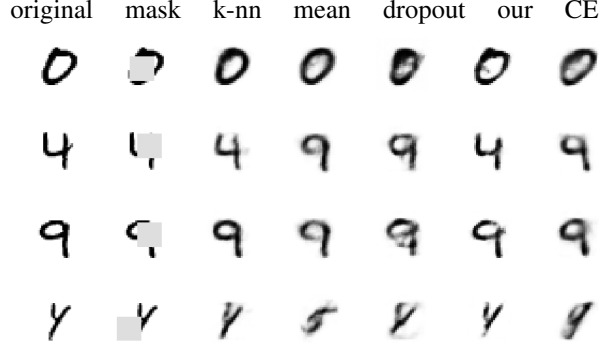

original    mask    k-nn    mean    dropout    our    CE

Figure 2: Reconstructions of partially incomplete images using the autoencoder. From left: (1) original image, (2) image with missing pixels passed to autoencooder; the output produced by autoencoder when unknown pixels were initially filled by (3) k-nn imputation and (4) mean imputation; (5) the results obtained by autoencoder with dropout, (6) our method and (7) context encoder. All columns except the last one were obtained with loss function computed based on pixels from outside the mask (no fully observable data available in training phase). It can be noticed that our method gives much sharper images than the competitive methods.

it is sufficient to prove that $T \in \mathcal{F}_w$. Let $M_p(r) = \max(0, r - p)$. Clearly

$$T_{p_0,p_1,p_2} = \frac{1}{p_1 - p_0} \cdot (M_{p_0} - M_{p_1}) - \frac{1}{p_2 - p_1} \cdot (M_{p_2} - M_{p_1}).$$

However, directly from (5) we see that $M_p \in \mathcal{F}_w$ for every $p$, and consequently $T$ and $Q$ are also in $\mathcal{F}_w$.

Now let us fix an arbitrary bounded continuous function $G$. We show that $G \in \mathcal{F}_w$. To observe this, take an arbitrary uniformly bounded sequence of piecewise linear functions described before which is convergent pointwise to $G$. By the Lebesgue dominated convergence theorem we obtain that $G \in \mathcal{F}_w$.

Therefore $\cos(\cdot), \sin(\cdot) \in \mathcal{F}_w$ holds consequently also for the function $e^{ir} = \cos r + i \sin r$ we have the equality

$$\int \exp(iw^T x) d\mu(x) = \int \exp(iw^T x) d\nu(x).$$

Since $w \in \mathbb{R}^D$ was chosen arbitrarily, this means that the characteristic functions of two measures coincide, and therefore $\mu = \nu$. ☐

It is possible to obtain an analogical result for RBF activation function. Moreover, we can also get more general result under stronger assumptions on considered probability measures. More precisely, if a given family of neurons satisfies UAP, then their generalization is also capable of identifying any probability measure with compact support. Complete analysis of both cases is presented in Supplementary Material (section 2).

## 5    Experiments

We evaluated our model on three types of architectures. First, as a proof of concept, we verified the proposed approach in the context of autoencoder (AE). Next we applied multilayer perceptron (MLP) to multiclass classification problem and finally we used shallow radial basis function network (RBFN) in binary classification. For a comparison we only considered methods with publicly available codes and thus many methods described in the related work section have not been taken into account. The code implementing the proposed method is available at `https://github.com/lstruski/Processing-of-missing-data-by-neural-networks`.

**Autoencoder.** Autoencoder (AE) is usually used for generating compressed representation of data. However, in this experiment, we were interested in restoring corrupted images, where part of data was hidden.

Table 1: Mean square error of reconstruction on MNIST incomplete images (we report the errors calculated over the whole area, inside and outside the mask). Described errors are obtained for images with intensities scaled to $[0, 1]$.

| | only missing data | | | | complete data |
|---|---|---|---|---|---|
| | **k-nn** | **mean** | **dropout** | **our** | **CE** |
| Total error | 0.01189 | 0.01727 | 0.01379 | **0.01056** | 0.01326 |
| Error inside the mask | 0.00722 | 0.00898 | 0.00882 | 0.00810 | **0.00710** |
| Error outside the mask | 0.00468 | 0.00829 | 0.00498 | **0.00246** | 0.00617 |

As a data set, we used grayscale handwritten digits retrieved from MNIST database. For each image of the size $28 \times 28 = 784$ pixels, we removed a square patch of the size[5] $13 \times 13$. The location of the patch was uniformly sampled for each image. AE used in the experiments consists of 5 hidden layers with 256, 128, 64, 128, 256 neurons in subsequent layers. The first layer was parametrized by ReLU activation functions, while the remaining units used sigmoids[6].

As describe in Section 1, our model assumes that there is no complete data in training phase. Therefore, the loss function was computed based only on pixels from outside the mask.

As a baseline, we considered combination of analogical architecture with popular imputation techniques:

*k-nn*: Missing features were replaced with mean values of those features computed from the $K$ nearest training samples (we used $K = 5$). Neighborhood was measured using Euclidean distance in the subspace of observed features.

*mean*: Missing features were replaced with mean values of those features computed for all (incomplete) training samples.

*dropout*: Input neurons with missing values were dropped[7].

Additionally, we used a type of context encoder (*CE*), where missing features were replaced with mean values, however in contrast to mean imputation, the complete data were used as an output of the network in training phase. This model was expected to perform better, because it used complete data in computing the network loss function.

Incomplete inputs and their reconstructions obtained with various approaches are presented in Figure 2 (more examples are included in Supplementary Material, section 3). It can be observed that our method gives sharper images then the competitive methods. In order to support the qualitative results, we calculated mean square error of reconstruction (see Table 1). Quantitative results confirm that our method has lower error than imputation methods, both inside and outside the mask. Moreover, it overcomes CE in case of the whole area and the area outside the mask. In case of the area inside the mask, CE error is only slightly better than ours, however CE requires complete data in training.

**Multilayer perceptron.** In this experiment, we considered a typical MLP architecture with 3 ReLU hidden layers. It was applied to multiclass classification problem on Epileptic Seizure Recognition data set (ESR) taken from [35]. Each 178-dimensional vector (out of 11500 samples) is EEG recording of a given person for 1 second, categorized into one of 5 classes. To generate missing attributes, we randomly removed 25%, 50%, 75% and 90% of values.

In addition to the imputation methods described in the previous experiment, we also used iterative filling of missing attributes using Multiple Imputation by Chained Equation (*mice*), where several imputations are drawing from the conditional distribution of data by Markov chain Monte Carlo techniques [10, 11]. Moreover, we considered the mixture of Gaussians (*gmm*), where missing

Table 2: Classification results on ESR data obtained using MLP (the results of CE are not bolded, because it had access to complete examples).

| | only missing data | | | | | | complete data |
|---|---|---|---|---|---|---|---|
| % of missing | k-nn | mice | mean | gmm | dropout | our | CE |
| 25% | 0.773 | **0.823** | 0.799 | **0.823** | 0.796 | 0.815 | 0.812 |
| 50% | 0.773 | 0.816 | 0.703 | 0.801 | 0.780 | **0.817** | 0.813 |
| 75% | 0.628 | 0.786 | 0.624 | 0.748 | 0.755 | **0.787** | 0.792 |
| 90% | 0.615 | 0.670 | 0.596 | 0.697 | 0.749 | **0.760** | 0.771 |

Table 3: Summary of data sets with internally absent attributes.

| Data set | #Instances | #Attributes | #Missing |
|---|---|---|---|
| bands | 539 | 19 | 5.38% |
| kidney disease | 400 | 24 | 10.54% |
| hepatitis | 155 | 19 | 5.67% |
| horse | 368 | 22 | 23.80% |
| mammographics | 961 | 5 | 3.37% |
| pima | 768 | 8 | 12.24% |
| winconsin | 699 | 9 | 0.25% |

features were replaced with values sampled from GMM estimated from incomplete data using EM algorithm[8].

We applied double 5-fold cross-validation procedure to report classification results and we tuned required hyper-parameters. The number of the mixture components for our method was selected in the inner cross-validation from the possible values $\{2, 3, 5\}$. Initial mixture of Gaussians was selected using classical GMM with diagonal matrices. The results were assessed using classical accuracy measure.

The results presented in Table 2 show the advantage of our model over classical imputation methods, which give reasonable results only for low number of missing values. It is also slightly better than dropout, which is more robust to the number of absent attributes than typical imputations. It can be seen that our method gives comparable scores to CE, even though CE had access to complete training data. We also ran MLP on complete ESR data (with no missing attributes), which gave 0.836 of accuracy.

**Radial basis function network.** RBFN can be considered as a minimal architecture implementing our model, which contains only one hidden layer. We used cross-entropy function applied on a softmax in the output layer. This network suits well for small low-dimensional data.

For the evaluation, we considered two-class data sets retrieved from UCI repository [36] with internally missing attributes, see Table 3 (more data sets are included in Supplementary Materials, section 4). Since the classification is binary, we extended baseline with two additional SVM kernel models which work directly with incomplete data without performing any imputations:

*geom*: Its objective function is based on the geometric interpretation of the margin and aims to maximize the margin of each sample in its own relevant subspace [26].

*karma*: This algorithm iteratively tunes kernel classifier under low-rank assumptions [29].

The above SVM methods were combined with RBF kernel function.

We applied analogical cross-validation procedure as before. The number of RBF units was selected in the inner cross-validation from the range $\{25, 50, 75, 100\}$. Initial centers of RBFNs were randomly selected from training data while variances were samples from $N(0, 1)$. For SVM methods, the margin parameter $C$ and kernel radius $\gamma$ were selected from $\{2^k : k = -5, -3, \ldots, 9\}$ for both parameters. For karma, additional parameter $\gamma_{karma}$ was selected from the set $\{1, 2\}$.

Table 4: Classification results obtained using RBFN (the results of CE are not bolded, because it had access to complete examples).

| data | only missing data | | | | | | | | complete data |
|------|-------|------|------|------|------|------|---------|------|------|
| | **karma** | **geom** | **k-nn** | **mice** | **mean** | **gmm** | **dropout** | **our** | **CE** |
| bands | 0.580 | 0.571 | 0.520 | 0.544 | 0.545 | 0.577 | **0.616** | 0.598 | 0.621 |
| kidney | **0.995** | 0.986 | 0.992 | 0.992 | 0.985 | 0.980 | 0.983 | 0.993 | 0.996 |
| hepatitis | 0.665 | 0.817 | 0.825 | 0.792 | 0.825 | 0.820 | 0.780 | **0.846** | 0.843 |
| horse | 0.826 | 0.822 | 0.807 | 0.820 | 0.793 | 0.818 | 0.823 | **0.864** | 0.858 |
| mammogr. | 0.773 | 0.815 | 0.822 | 0.825 | 0.819 | 0.803 | 0.814 | **0.831** | 0.822 |
| pima | 0.768 | 0.766 | 0.767 | **0.769** | 0.760 | 0.742 | 0.754 | 0.747 | 0.743 |
| winconsin | 0.958 | 0.958 | 0.967 | **0.970** | 0.965 | 0.957 | 0.964 | **0.970** | 0.968 |

The results, presented in Table 4, indicate that our model outperformed imputation techniques in almost all cases. It partially confirms that the use of raw incomplete data in neural networks is usually better approach than filling missing attributes before learning process. Moreover, it obtained more accurate results than modified kernel methods, which directly work on incomplete data.

## 6 Conclusion

In this paper, we proposed a general approach for adapting neural networks to process incomplete data, which is able to train on data set containing only incomplete samples. Our strategy introduces input layer for processing missing data, which can be used for a wide range of networks and does not require their extensive modifications. Thanks to representing incomplete data with probability density function, it is possible to determine more generalized and accurate response (activation) of the neuron. We showed that this generalization is justified from a theoretical perspective. The experiments confirm its practical usefulness in various tasks and for diverse network architectures. In particular, it gives comparable results to the methods, which require complete data in training.

## Acknowledgement

This work was partially supported by National Science Centre, Poland (grants no. 2016/21/D/ST6/00980, 2015/19/B/ST6/01819, 2015/19/D/ST6/01215, 2015/19/D/ST6/01472). We would like to thank the anonymous reviewers for their valuable comments on our paper.

## Footnotes

[1]More precisely, $F_S$ equals a density $F$ conditioned on the observed attributes.

[2]An example of degenerate density is a degenerate Gaussian $N(m, \Sigma)$, for which $\Sigma$ is not invertible. A degenerate Gaussian is defined on affine subspace (given by image of $\Sigma$), see [34] for details. For simplicity we use the same notation $N(m, \Sigma)$ to denote both standard and degenerate Gaussians.

[3]One can show that the conditional density of a missing point $(x, J)$ sufficiently distant from the data reduces to only one Gaussian, which center is nearest in the Mahalanobis distance to $\text{Aff}[x, J]$

[4]If huge amount of complete data is available during training, one should use variants of EM algorithm to estimate data density. It could be either used directly as a missing data density or tuned by neural networks with small amount of missing data.

[5]In the case when the removed patch size was smaller, all considered methods performed very well and cannot be visibly distinguished.

[6]We also experimented with ReLU in remaining layers (except the last one), however the results we have obtained were less plausible.

[7]Values of the remaining neurons were divided by $1 - dropout\ rate$

[8]Due to the high-dimensionality of MNIST data, mice was not able to construct imputations in previous experiment. Analogically, EM algorithm was not able to fit GMM because of singularity of covariance matrices.

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
