[Supplementary Material · supplementary_material.pdf]

# Processing of missing data by neural networks
# Supplementary Material

**Marek Śmieja**
marek.smieja@uj.edu.pl

**Łukasz Struski**
lukasz.struski@uj.edu.pl

**Jacek Tabor**
jacek.tabor@uj.edu.pl

**Bartosz Zieliński**
bartosz.zielinski@uj.edu.pl

**Przemysław Spurek**
przemyslaw.spurek@uj.edu.pl

Faculty of Mathematics and Computer Science
Jagiellonian University
Łojasiewicza 6, 30-348 Kraków, Poland

## 1 Missing data representation

In this section, we show how to regularize typical conditional probability density function. Next, we present complete formulas for a conditional density in the case of the mixture of Gaussians.

### 1.1 Regularized conditional density

Let us recall a definition of conditional density representing missing data points formulated in the paper. We assume that $F$ is a probability density function on data space $\mathbb{R}^D$. A missing data point $(x, J)$ can be represented by restricting $F$ to the affine subspace $S = \mathrm{Aff}[x, J]$, which gives a conditional density $F_S : S \to \mathbb{R}$ given by:

$$F_S(x) = \begin{cases} \frac{1}{\int_S F(s)ds} F(x), \text{ for } x \in S, \\ 0, \text{ otherwise,} \end{cases} \tag{1}$$

The natural choice for missing data density $F$ is to apply GMM. However, the straightforward application of GMM may lead to some practical problems with taking the conditional density (1). Thus, to provide better representation of missing data points we introduce additional regularization described in this section.

Let us observe that the formula (1) is not well-defined in the case when the density function $F$ is identically zero on the affine space $S = \mathrm{Aff}[x, J]$. In practice, the same problem appears numerically for the mixture of gaussians, because every component has exponentially fast decrease. In consequence, (1) either gives no sense, or trivializes[1] (reduces to only one gaussian) for points sufficiently far from the main clusters. To some extent we can also explain this problem with the fact that the real density can have much slower decrease to infinity than gaussians, and therefore the estimation of conditional density based on gaussian mixture becomes unreliable.

To overcome this problem, which occurs in the case of gaussian distributions, we introduce the regularized $\gamma$-conditional densities, where $\gamma > 0$ is a regularization parameter. Intuitively, the regularization allows to control the influence of $F$ outside $S = \mathrm{Aff}[x, J]$ on conditional density $F_S$. In consequence, the mixture components (in the case of GMM) could have higher impact on the final conditional density even if they are located far from $S$. The Figure 1 illustrates the regularization effect for different values of $\gamma$. We are indebted to the classical idea behind the definition of conditional probability.

| (a) Conditional density ($\gamma \to 0$) | (b) Intermediate case ($\gamma = 1$) | (c) Marginal density ($\gamma \to \infty$) |

Figure 1: Illustration of probabilistic representation $F_S$ of missing data point $(*, -1) \in \mathbb{R}^2$ for different regularization parameters $\gamma$ when data density is given by the mixture of two Gaussians.

Let $\gamma > 0$ be a regularization parameter. By regularized $\gamma$-restriction of $F$ to the affine subspace $S$ of $\mathbb{R}^D$ we understand

$$F^\gamma|_S(x) = \begin{cases} \int_{S_\perp^x} F(s) \cdot N(x, \gamma I_{S-x})(s)ds, & \text{if } x \in S, \\ 0 & \text{otherwise,} \end{cases}$$

where $S_\perp^x = \{w : (w - x) \perp (S - x)\}$ is the affine space consisting of all points which are at $x$ perpendicular to $S$, and $N(x, \gamma I_{S-x})$ is the degenerate normal density which has mean at $x$, is supported on $S$ and its covariance matrix is a rescaled identity (restricted to $S - x$). Then the regularized $\gamma$-conditional density $F_S^\gamma$ is defined as the normalization of $F^\gamma|_S$:

$$F_S^\gamma = \begin{cases} \frac{1}{\int_S F^\gamma|_S(s)ds} F^\gamma|_S & \text{for } s \in S, \\ 0 & \text{otherwise.} \end{cases}$$

The regularized density $F_S^\gamma$ has the following properties:

1. $F_S^\gamma$ is well-defined degenerate density on $S$ for every $\gamma$,
2. $F_S^\gamma$ converges to the conditional density $F_S$ with $\gamma \to 0$, see Figure 1(a)
3. $F_S^\gamma$ converges to the marginal density as $\gamma \to \infty$, see Figure 1(c).

One can easily see that the first point follows directly from the fact that $F$ is integrable. Since for an arbitrary function $g$ we have $\int g(s)N(x, \gamma I)(s)ds \to f(x)$, as $\gamma \to 0$, we obtain that

$$\lim_{\gamma \to 0} F^\gamma|_S(x) = F(x) \text{ for } x \in S.$$

Thus $F^\gamma|_S \to F|_S$, as $\gamma \to 0$. Analogously

$$\lim_{\gamma \to \infty} \int \frac{1}{N(0, \gamma I)(0)} g(s)N(x, \gamma I)(s)ds$$

$$= \lim_{\gamma \to \infty} \int g(s) \exp(-\frac{1}{2\gamma}\|x - s\|^2)ds = \int g(s)ds,$$

which implies that for large $\gamma$ the function $F_S^\gamma$ as a renormalization of $F^\gamma|_S$ at point $x$ converges to

$$\int_{S_\perp^x} F(s)ds,$$

which is exactly the value of marginal density at point $x$.

## 1.2 Gaussian model for missing data density

We consider the case of $F$ given by GMM and calculate analytical formula for the regularized $\gamma$-conditional density. To reduce the number of parameters and provide more reliable estimation in high dimensional space, we use diagonal covariance matrix for each mixture component.

We will need the following notation: given a point $x \in \mathbb{R}^D$ and a set of indexes $K \subset \{1, \ldots, D\}$ by $x_K$ we denote the restriction of $x$ to the set of indexes $K$. The complementary set to $K$ is denoted by $K'$. Given $x, y$, by $[x_{K'}, y_K]$ we denote a point in $\mathbb{R}^D$ which coordinates equal $x$ on $K'$ and $y$ on $K$. We use analogous notation for matrices.

One obtains the following exact formula for regularized restriction of gaussian density with diagonal covariance:

**Proposition 1.1.** *Let $N(m, \Sigma)$ be non-degenerate normal density with a diagonal covariance $\Sigma = \mathrm{diag}(\sigma_1, \ldots, \sigma_D)$. We consider a missing data point $(x, J)$ represented by the affine subspace $S = \mathrm{Aff}[x, J]$. Let $\gamma > 0$ be a regularization parameter.*

*The $\gamma$-regularized restriction of $F$ to $S$ at point $s = [x_{J'}, y_J] \in S$ equals:*

$$F^\gamma|_S(s) = C^\gamma_{m,\Sigma,S} N(m_S, \Sigma_S)(s),$$

*where*

$$m_S = [x_{J'}, m_J], \Sigma_S = [0_{J'J'}, \Sigma_{JJ}],$$

$$C^\gamma_{m,\Sigma,S} = \frac{1}{(2\pi)^{(D-|J|)/2} \prod_{l \in J'} (\gamma + \sigma_l)^{1/2}}$$

$$\cdot \exp(-\tfrac{1}{2} \sum_{l \in J'} \tfrac{1}{\gamma + \sigma_l} (m_l - x_l)^2).$$

Finally, by using the above proposition (after normalization) we get the formula for the regularized conditional density in the case of the mixture of gaussians:

**Corollary 1.1.** *Let $F$ be the mixture of nondegenerate gaussians*

$$F = \sum_i p_i N(m_i, \Sigma_i),$$

*where all $\Sigma_i = \mathrm{diag}(\sigma_1^i, \ldots, \sigma_D^i)$ and let $S = \mathrm{Aff}[x, J]$.*

*Then*

$$F^\gamma_S = \sum_i r_i N(m_S^i, \Sigma_S^i),$$

*where*

$$m_S^i = [x_{J'}, (m_i)_J], \Sigma_S^i = [0_{J'J'}, (\Sigma_i)_{JJ}],$$

$$r_i = \frac{q_i}{\sum_j q_j}, q_i = C^\gamma_{m_i, \Sigma_i, S} \cdot p_i,$$

$$C^\gamma_{m,\Sigma,S} = \frac{1}{(2\pi)^{(D-|J|)/2} \prod_{l \in J'} (\gamma + \sigma_l)^{1/2}}$$

$$\cdot \exp(-\tfrac{1}{2} \sum_{l \in J'} \tfrac{1}{\gamma + \sigma_l} (m_l - x_l^2)).$$

## 2 Theoretical analysis

In this section, we continue a theoretical analysis of our model. First, we consider a special case of RBF neurons for arbitrary probability measures. Next, we restrict our attention to the measures with compact supports and show that the identification property holds for neurons satisfying UAP

### 2.1 Identification property for RBF

RBF function is given by

$$\mathrm{RBF}_{m,\Sigma}(x) = N(m, \Sigma)(x),$$

where $m$ is an arbitrary point and $\Sigma$ is positively defined symmetric matrix. In some cases one often restricts to either diagonal or rescaled identities $\Sigma = \alpha I$, where $\alpha > 0$. In the last case we use the notation $\mathrm{RBF}_{m,\alpha}$ for $\mathrm{RBF}_{m,\alpha I}$.

**Theorem 2.1.** *Let $\mu, \nu$ be probabilistic measures. If*

$$\mathrm{RBF}_{m,\alpha}(\mu) = \mathrm{RBF}_{m,\alpha}(\nu) \text{ for every } m \in \mathbb{R}^D, \alpha > 0,$$

*then $\nu = \mu$.*

*Proof.* We will show that $\mu$ and $\nu$ coincide on every cube. Recall that $(\eta * f)(x) = \int \eta(y) \cdot f(x - y) d\lambda_N(x)$.

Let us first observe that for an arbitrary cube $K = a + [0, h]^D$

$$\int \mathbb{1}_K * N(0, \alpha) d\mu(x) = \int \mathbb{1}_K * N(0, \alpha) d\nu(x),$$

where $h > 0$ is arbitrary. This follows from the obvious observation that

$$\frac{1}{n^K} \sum_{i \in \mathbb{Z}^D \cap [0,n]^D} N(a + \tfrac{i}{n} h, \alpha I)$$

converges uniformly to $\mathbb{1}_K * N(0, \alpha I)$, as $n$ goes to $\infty$.

Since $\mathbb{1}_{K + \frac{1}{n}[0,1]^n} * N(0, \frac{1}{n^4} I)$ converges pointwise to $\mathbb{1}_K$, analogously as before by applying Lebesgue dominated convergence theorem we obtain the assertion. $\square$

## 2.2 General identification property

We begin with recalling the UAP (universal approximation property). We say that a family of neurons $\mathcal{N}$ has UAP if for every compact set $K \subset \mathbb{R}^D$ and a continuous function $f : K \to \mathbb{R}$ the function $f$ can be arbitrarily close approximated with respect to supremum norm by $\mathrm{span}(\mathcal{N})$ (linear combinations of elements of $\mathcal{N}$).

Our result shows that if a given family of neurons satisfies UAP, then their generalization allows to distinguish any two probability measures with compact support:

**Theorem 2.2.** *Let $\mu, \nu$ be probabilistic measures with compact support. Let $\mathcal{N}$ be a family of functions having UAP.*

*If*

$$n(\mu) = n(\nu) \text{ for every } n \in \mathcal{N}, \tag{2}$$

*then $\nu = \mu$.*

*Proof.* Since $\mu, \nu$ have compact support, we can take $R > 1$ such that $\mathrm{supp}\,\mu, \mathrm{supp}\,\nu \subset B(0, R-1)$, where $B(a, r)$ denotes the closed ball centered at $a$ and with radius $r$. To prove that measures $\mu, \nu$ are equal it is obviously sufficient to prove that they coincide on each ball $B(a, r)$ with arbitrary $a \in B(0, R-1)$ and radius $r < 1$.

Let $\phi_n$ be defined by

$$\phi_n(x) = 1 - n \cdot d(x, B(a, r)) \text{ for } x \in \mathbb{R}^D,$$

where $d(x, U)$ denotes the distance of point $x$ from the set $U$. Observe that $\phi_n$ is a continuous function which is one on $B(a, r)$ an and zero on $\mathbb{R}^D \setminus B(a, r + 1/n)$, and therefore $\phi_n$ is a uniformly bounded sequence of functions which converges pointwise to the characteristic funtion $\mathbb{1}_{B(a,r)}$ of the set $B(a, r)$.

By the UAP property we choose $\psi_n \in \mathrm{span}(\mathcal{N})$ such that

$$\sup_{x \in B(0,R)} |\phi_n(x) - \psi_n(x)| \leq 1/n.$$

By the above also $\psi_n$ restricted to $B(0, R)$ is a uniformly bounded sequence of functions which converges pointwise to $\mathbb{1}_{B(a,r)}$. Since $\psi_n \in \mathcal{N}$, by (2) we get

$$\int \psi_n(x) d\mu(x) = \int \psi_n(x) d\nu(x).$$

original   mask   k-nn   mean   dropout   our   CE

Figure 2: More reconstructions of partially incomplete images using the autoencoder. From left:
(1) original image, (2) image with missing pixels passed to autoencooder; the output produced by
autoencoder when absent pixels were initially filled by (3) k-nn imputation and (4) mean imputation;
(5) the results obtained by autoencoder with (5) dropout, (6) our method and (7) context encoder. All
columns except the last one were obtained with loss function computed based on pixels from outside
the mask (no fully observable data available in training phase). It can be noticed that our method
gives much sharper images then the competitive methods.

Now by the Lebesgue dominated convergence theorem we trivially get

$$\int \psi_n(x)d\mu(x) = \int_{B(0,R)} \psi_n(x)d\mu(x) \to \mu(B(a,r)),$$
$$\int \psi_n(x)d\nu(x) = \int_{B(0,R)} \psi_n(x)d\nu(x) \to \nu(B(a,r)),$$

which makes the proof complete. $\square$

## 3   Reconstruction of incomplete MNIST images

Due to the limited space in the paper, we could only present 4 sample images from MNIST experiment.
In Figure 2, we present more examples from this experiment.

Table 1: Summary of data sets, where 50% of values were removed randomly.

| Data set | #Instances | #Attributes |
|---|---|---|
| australian | 690 | 14 |
| bank | 1372 | 4 |
| breast cancer | 699 | 8 |
| crashes | 540 | 20 |
| diabetes | 768 | 8 |
| fourclass | 862 | 2 |
| heart | 270 | 13 |
| liver disorders | 345 | 6 |

Table 2: Classification results measured by accuracy on UCI data sets with 50% of removed attributes.

| data | karma | geom | k-nn | mice | mean | dropout | our |
|---|---|---|---|---|---|---|---|
| australian | **0.833** | 0.802 | 0.820 | 0.826 | 0.808 | 0.812 | **0.833** |
| bank | **0.799** | 0.740 | 0.763 | 0.793 | 0.788 | 0.722 | 0.795 |
| breast cancer | 0.938 | 0.874 | 0.902 | 0.942 | 0.938 | 0.911 | **0.951** |
| crashes | **0.920** | 0.914 | 0.898 | 0.894 | 0.892 | 0.900 | **0.920** |
| diabetes | 0.695 | 0.644 | 0.673 | **0.708** | 0.699 | 0.675 | 0.690 |
| fourclass | **0.808** | 0.653 | 0.766 | 0.776 | 0.766 | 0.731 | 0.737 |
| heart | 0.755 | 0.738 | 0.725 | 0.751 | 0.725 | 0.722 | **0.770** |
| liver disorders | 0.530 | 0.591 | 0.565 | 0.576 | 0.562 | 0.571 | **0.608** |

## 4 Additional RBFN experiment

In addition to data sets reported in the paper, we also ran RBFN on 8 examples retrieved from UCI repository, see Table 1. These are complete data sets (with no missing attributes). To generate missing samples, we randomly removed 50% of values.

The results presented in Table 2 confirm the effects reported in the paper. Our method outperformed imputation techniques in almost all case and was slightly better than karma algorithm.

## 5 Computational complexity

We analyze the computational complexity of applying a layer for missing data processing with $k$ Gaussians for modeling missing data density. Given an incomplete data point $S = \mathrm{Aff}[x, J]$, where $x \in \mathbb{R}^D$ and $J \subset \{1, \ldots, N\}$, the cost of calculation of regularized (degenerate) density $F_S^\gamma$ is $O(k|J'|)$, where $J' = \{1, \ldots, N\} \setminus J$ (see Corollary 1.1. in supplementary material). Computation of a generalized ReLU activation (Theorem 3.1) takes $O(kD + k|J|)$. If we have $t$ neurons in the first layer, then a total cost of applying our layer is $O(k|J'| + tk(D + |J|))$.

In contrast, for a complete data point we need to compute $t$ ReLU activations, which is $O(tD)$. In consequence, generalized activations can be about $2k$ times slower than working on complete data.

## 6 Learning missing data density

To run our model we need to define initial mixture of Gaussians. This distribution is passed to the network and its parameters are tuned jointly with remaining network weights to minimize the overall cost of the network.

We illustrate this behavior on the following toy example. We generated a data set from the mixture of four Gaussians; two of them were labeled as class 1 (green) while the remaining two were labeled as class 2 (blue), see Figure 3(a). We removed one of the attributes from randomly selected data points $x = (x_1, x_2)$ with $x_1 < 0$. In other words, we generated missing samples only from two Gaussian on the left. Figure 3(b) shows initial GMM passed to the network. As can be seen this GMM matches neither a data density nor a density of missing samples. After training, we get a GMM, where its first component estimates a density of class 1, while the second component matches class 2, see Figure 3(c). In consequence, learning missing data density by the network helped to perform better classification than estimating GMM directly by EM algorithm.

(a) Reference classification

(b) Initial Gaussians

(c) Final Gaussians and resulted classification

Figure 3: Toy example of learning missing data density by the network.

## Footnotes

[1]One can show that the conditional density of a missing point $(x, J)$ sufficiently distant from the data reduces to only one gaussian, which center is nearest in the Mahalanobis distance to $\mathrm{Aff}[x, J]$