[Reviews · NeurIPS 2018]

Reviewer 1



The authors introduce

Reviewer 2



The paper provides a theoretical and practical justification on using a density function to represent missing data while training a neural networks. An obvious upside is that training can be done with incomplete data, unlike denoising autoencoder for example; this can be very helpful in many applications. My comments are: - It is stated that if all the attributes are complete then the density is not used; if we have access to a huge amount of complete training data and relatively small amount of training missing data, how trustworthy is our estimation of density function? Can’t we benefit from the complete data? Do we really have to remove attributes as is done in ESR task? - In the above case, would denoising autoencoder outperform? - How would the generalized activation impact the training time? Any observation or input in this regard can be helpful to better assess the usefulness of this method. - With regards to the experiments, It would have been much better if some experiments were carried out to compare the proposed method with one of [2,3,4] when complete data is available for training. Also, some results showing performance vs amount of missing data (e.g. 75%, 50% and 25%) could give a helpful insight. - Please also look at the following paper and see if a fair comparison can be made. GAIN: Missing Data Imputation using Generative Adversarial Nets - The second paragraph in introduction : ‘be understand’ -> be understood - Section 4 first paragraph: ‘lose’ -> loss - In section 3, please define N. - There are some old and nice works on dealing with NN and missing data that could be cited. ### Thanks for the reply. I am happy with the responses. With regard to the GAIN paper, what I meant was to try to conceptually compare your method with that and discuss the differences or similarities.

Reviewer 3



The authors introduce a new NN layer capable of processing missing data. At the core of the new layer lies a generalized neuron activation function that operates over probability densities. To estimate the densities the authors consider other points in an input dataset that lie in the subspace of known coordinates of an example and apply a mixture of Gaussians model. What I find most interesting in this framework is the generalized activation function. Given the assumptions presented in the paper the analysis seems straightforward. The authors evaluate their method against a diverse array of baselines and demonstrate that it performs favorably compared to those.